# Use of autobiographical stimuli as a mood manipulation procedure: Systematic mapping review

**Dolores Fernández-Pérez**[1,2,3], **Abel Toledano-González**[2,3]*, **Laura Ros**[1,2,3], **José M. Latorre**[1,2,3]

**1** Department of Psychology, Faculty of Medicine, University of Castilla-La Mancha, Albacete, Spain, **2** Department of Psychology, Faculty of Health Sciences, University of Castilla-La Mancha, Talavera de la Reina, Spain, **3** Neurological Disabilities Research Institute, Albacete, Spain

* atoledanogonzalez@gmail.com

## Abstract

### Background

In recent years, mood induction procedures have been developed in experimental settings that are designed to facilitate studying the impact of mood states on biological and psychological processes. The aim of the present study was to conduct a systematic mapping review with the intention of describing the state of the art in the use of different types of autobiographical stimuli for mood induction procedures.

### Methods

Based on a search for publications from the period 2000–2021, conducted in four recognised databases (Scopus, Medline (PubMed), PsycINFO and Web of Science), we analysed a total of 126 published articles. Text mining techniques were used to extract the main themes related.

### Results

The induction of emotions through autobiographical memories is an area under construction and of growing interest. The data mining approach yielded information about the main types of stimuli used in these procedures, highlighting those that only employ a single type of cue, as well as the preference for verbal cues over others such as musical, olfactory and visual cues. This type of procedure has been used to induce both positive and negative emotions through tasks that require access to personal memories of specific events from a cue, requiring the person to set in motion different cognitive processes. The use of the latest technologies (fMRI, EEG, etc.) is also shown, demonstrating that this is a cutting-edge field of study.

**Data Availability Statement:** As this article is a review, the results of the final articles are extracted after screening. They can be found in the table or

supplementary material as well as in the corresponding section.

**Funding:** The authors received funding from the European Regional Development Funds (FEDER) [2018/11744] [2019/7375] [2020/3771]. This work was supported by the Castilla-La Mancha Department of Education, Culture and Sports and the European Regional Development Fund under SPBLY/19/180501/000181 grant and the Spanish Ministry of Science, Innovation and Universities under PID2019-103956RB-I00 grant. The funders had no role in the study design, data collection and analysis, decision to publish, or preparation of the manuscript.

**Competing interests:** The authors have declared that no competing interests exist

## Conclusions

Despite the study of mood induction procedures still being a growing field, the present review provides a novel overview of the current state of the art in the field, which may serve as a framework for future studies on the topic.

## 1. Introduction

In recent years, there has been growing interest in studying emotions and their impact on different psychological processes and characteristics (e.g., [1, 2]). The study of these relationships in natural real-life scenarios is highly complex, which has led to the development of various mood induction techniques and procedures that are able to elicit emotional experiences in experimental situations. These procedures have now emerged as the most rigorous method of testing the causal effect of emotions on both psychological and biological variables [3] Mood Induction Procedures (MIPs) are defined as artificial and controlled strategies whose aim is to momentarily change an individual's mood, in a way that generates emotions that are equivalent to naturally occurring moods [4] One of the most widely used procedures in mood induction is that of accessing Autobiographical Memories (AMs).

AM refers to memories of personally experienced events. It can be understood as a hierarchical network extending from general memories, with semantic and conceptual content, to specific memories, situated in space and time, with a large number of sensory-perceptual details [5] AMs involve accessing personal events with an emotional load, such that evoking them allows the emotions experienced in the original event to be reactivated [6] Indeed, the access to AMs activates similar areas of the brain as those activated in response to the emotional experience involved in the event recalled [7, 8] which corroborates the validity of using such techniques as part of MIPs.

Emotion induction through autobiographical recall is considered effective in inducing certain basic emotions, such as anger, happiness, fear, disgust and sadness [3] having proven to be more effective than other techniques such as guided imagination [9, 10] In MIPs using autobiographical recall, participants are typically asked to access an event from their past as vividly as possible, one in which they felt an emotion such as happiness, sadness or anger, reliving sensations, perceptions and responses, in order to reactivate that same emotion [11, 12]. When retrieving AMs, individuals may begin from a general, abstract level (depending on the significance of the cue) and then reach a more specific, detailed level of memory. However, when much of the information concerning a memory is already present in the cue (e.g. a personal photograph), participants can immediately access the specific event [13, 14]. Hence, different types of stimuli are sometimes used (e.g. images, colours, songs, words, etc.) to facilitate access to the content of the AM. Additionally, the more accessible and significant the memories are, the more effective is the procedure and the manipulation of mood [15]. This is especially important in the case of older adults with greater difficulties in retrieving AMs or persons presenting depressive symptoms [16], with whom the use of powerful cues may facilitate memory retrieval and trigger a faster effect.

The study of MIPs is especially interesting given their implications from a research perspective, as the starting point from which to study the role of emotional processes in behavioural and cognitive mood disorders, and from a clinical viewpoint, where they can be applied in treating certain negative psychological states, such as mood disorders associated with problems of emotion regulation [17]. Specifically, the use of MIPs through autobiographical recall is

particularly interesting, on the one hand, for its effectiveness in the regulation of emotions, being able to reduce negative emotional states and increase the positive ones [18]. On the other hand, research on emotional processes has been surrounded by methodological problems as it has been carried out in laboratory environments where it is difficult to compare the induced emotion with that experienced in the natural environment, due to the lack of intensity and experiential content [19]. Given this fact, the potential of MIPs based on access to autobiographical memories lies in their validity from an ecological point of view, since the retrieval of past events through autobiographical stimuli is a frequent cause of emotional states in everyday life [20]. Thus, the use of this type of procedures helps to increase the understanding of emotional processes and thus contribute to the advancement of interventions aimed at improving the psychological well-being of individuals.

To date, a substantial number of articles have been published whose specific aim is to assess MIPs that use autobiographical stimuli, as well as many in which autobiographical recall MIPs are not a specific aim but are part of the experimental methodology. In view of this diversity, it is interesting to conduct a review that goes beyond a concrete topic, as is the case of theoretical reviews, and which can provide an overview of the state of the art in this field. For this reason, in the present review, it is our intention to establish links by means of a conceptual map, which may help us organise and visually present the knowledge thus far available on mood induction procedures that use different types of autobiographical stimuli, revealing the trends and gaps in the literature that could represent future research opportunities.

The aim of this study is to describe the state of the art in the use of different types of autobiographical stimuli in emotion induction. To this end, we follow a systematic mapping review approach to provide an overview of experimental studies on mood induction using autobiographical stimuli, with the hope of revealing patterns that are not evident at first sight.

## 2. Materials and methods

In the present work, a mapping review has been followed since this methodology offers a visual synthesis of the state of a subject in which there is an abundance and above all diversity of research and, since it serves as a preliminary step for a subsequent systematic review, it also helps to find possible gaps and therefore future lines of research [21].

To conduct the present review, we followed the steps outlined in the *Template for a Mapping Study Protocol* [22], undertaking the three phases proposed: 1) Identification of the research guidelines (defining and identifying the dimensions to be analysed and the research); 2) Data collection (in accordance with the inclusion and exclusion criteria) and 3) Results. A checklist in line with the *Template for a Mapping Study Protocol* is provided in the supplementary material (S1 File).

### 2.1. Identification of research guidelines

The research questions (RQs) for this review were as follows:

RQ1: How many articles on emotion induction through autobiographical stimuli have been published in the last twenty years?

RQ2: Where have the studies on emotion induction interventions using autobiographical stimuli been published? In what areas of psychology and related disciplines can we find studies on emotion induction using autobiographical stimuli?

RQ3: Is this type of procedure a specialised field?

RQ4: What are the re-occurring topics in the use of autobiographical stimuli for emotion induction?

RQ5: What types of autobiographical stimuli are most frequently used in MIPs?

RQ6: What types of emotions are most frequently elicited using procedures based on AM recall?

## 2.2. Data collection

**2.2.1. Search process.**   To find relevant studies, we consulted the primary databases related to the area of study, these being Scopus (Elsevier), Medline (PubMed), PsycINFO (American Psychological Association) and Web Of Science. A customised search string was used, as follows: ("Autobiographical Memory") AND ("Emotional Induction" OR "Emotion Regulation" OR "Emotion"), using the complete string in the fields of title, abstract and keywords.

**2.2.2. Inclusion and exclusion criteria.**   The inclusion and exclusion criteria were the following:

Inclusion criteria:

- Articles on experimental studies that used autobiographical stimuli as a method of mood induction (e.g.: cross-sectional studies, quasi-experimental studies, randomized controlled trial studies).

- Articles published between 2000 and 2021 (with June as the last complete month of the search). It was decided to start from 2000, as the non-digitalization of most of the articles published before that date could bias our results. Thus, we ensured access to a more reliable number of articles.

- Articles including examples of young (17 to 35 years), middle-aged (35 to 60 years) and older adults (over 60 years).

Exclusion criteria:

- Articles not published in English or Spanish.

- Articles on systematic reviews (including meta-analyses). These were taken into account, however, in order not to miss references).

- Grey literature, including editorials, extended abstracts, symposia, book chapters, technical reports, case studies, etc.

- Articles unrelated to research with human participants.

- Articles using pharmacological treatments.

**2.2.3. Selection process.**   The previously described search string was implemented in the different databases selected. The search results were uploaded to Covidence (a web-based tool for managing systematic review that aims to make evidence synthesis a more proficient process) [23] and duplicate articles were automatically eliminated. Fig 1 shows the flow diagram for the selection process described below.

After eliminating any duplicates (n = 1426), a total of 3296 articles were obtained. Two researchers independently reviewed and filtered all 3296 articles, reading and analysing the titles, abstract and keywords and following the inclusion and exclusion criteria. When the two researchers were unable to agree, any discrepancies were resolved by a third colleague. This

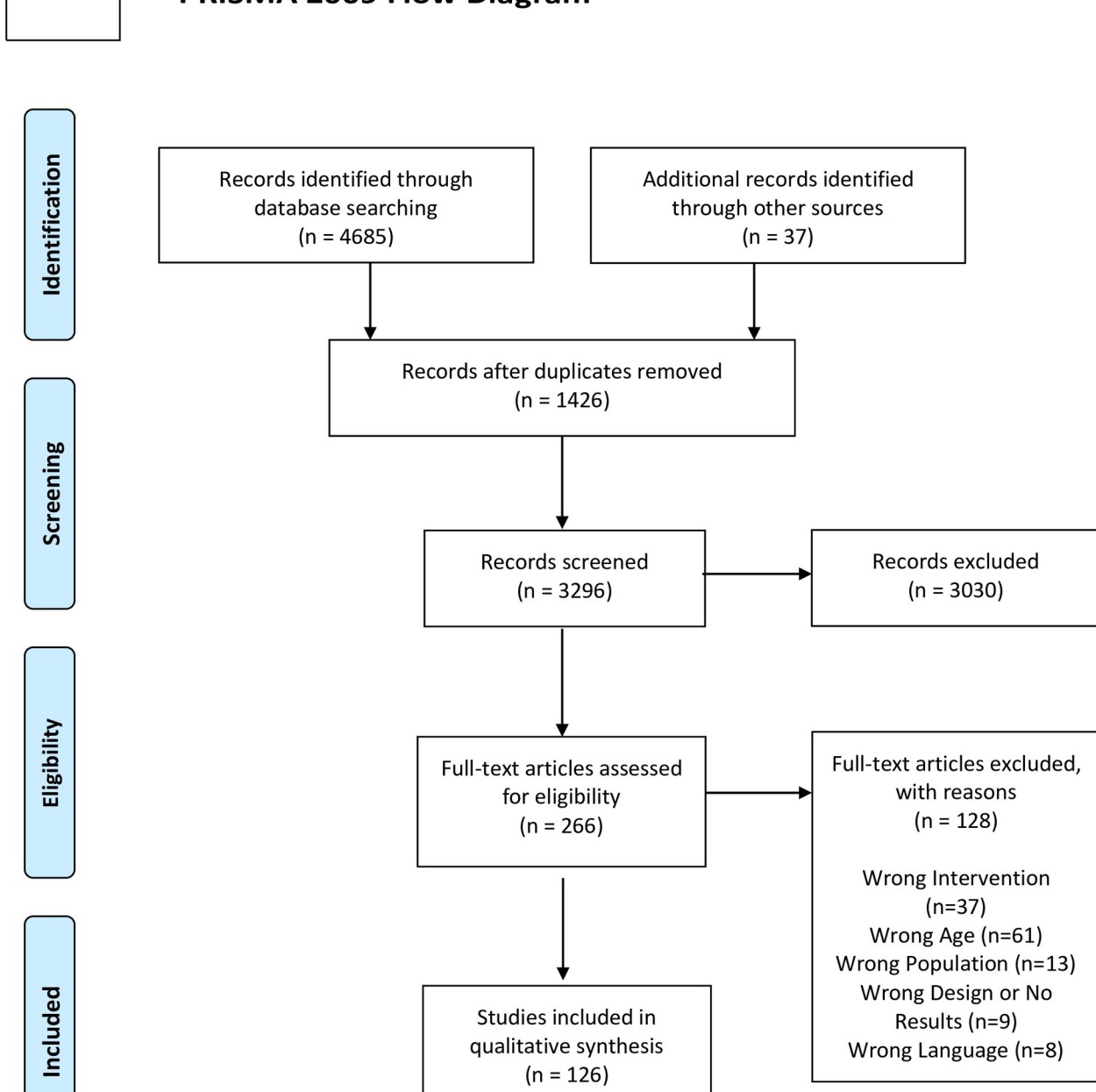

**PRISMA 2009 Flow Diagram**

**Identification**

Records identified through database searching
(n = 4685)

Additional records identified through other sources
(n = 37)

**Screening**

Records after duplicates removed
(n = 1426)

Records screened
(n = 3296)

Records excluded
(n = 3030)

**Eligibility**

Full-text articles assessed for eligibility
(n = 266)

Full-text articles excluded, with reasons
(n = 128)

Wrong Intervention (n=37)
Wrong Age (n=61)
Wrong Population (n=13)
Wrong Design or No Results (n=9)
Wrong Language (n=8)

**Included**

Studies included in qualitative synthesis
(n = 126)

*From:* Moher D, Liberati A, Tetzlaff J, Altman DG, The PRISMA Group (2009). *P*referred *R*eporting *I*tems for *S*ystematic Reviews and *M*eta-*A*nalyses: The PRISMA Statement. PLoS Med 6(7): e1000097. doi:10.1371/journal.pmed1000097

For more information, visit www.prisma-statement.org.

**Fig 1. Prisma flow diagram.**

first selection phase yielded 266 articles, which were then filtered again by the same two researchers, who independently read and analysed the complete content of each article. The reasons for articles eliminated at this stage (n = 128) were as follows: 1) Wrong Intervention (n = 37); 2) Wrong Age (n = 61); 3) Wrong Population (n = 13); 4) Wrong Design or No Results (n = 9); 5) Wrong Language (n = 8). As in the previous phase, any disagreements on whether to exclude or maintain a particular article were resolved independently by two independent researchers. This resulted in a total of 138 articles, which were subjected to a quality analysis, with 12 being dropped from the sample. We analysed a total of 126 published articles, of which 39 were randomized controlled trials, 78 were quasi-experimental studies and 9 were cross-sectional studies. The references from all the articles are available in Supplementary Material 2 (S2 File).

Two researchers (DFP and ATG) conducted the search of the literature. Two researchers (DFP and ATG) reviewed the relevance of the titles and abstracts according to the established inclusion and exclusion criteria. Data extraction was carried out by two researchers (DFP and ATG). Any discrepancies were resolved by a third colleague (JML or LR). A data extraction form and an abstract extraction form were designed for subsequent analysis (see S3 File).

**2.2.4. Quality analysis.** The quality of the studies and the presence of bias were determined using the Joanna Briggs Institute Critical Appraisal Checklist (JBC) [24], which was implemented independently by two researchers. This tool of proven validity allows us to analyse the included articles by determining their quality according to the type of design. We used the appraisal checklists for cross-sectional studies, quasi-experimental studies and randomised controlled trials, with 8, 9 and 11 items, respectively, each of which is marked "Yes", "No", "Unclear" or "Not Applicable". Although these are qualitative appraisal tools to be applied by reviewers, for the present study and with the aim of determining the quality of the studies and their inclusion or exclusion, we established a point-based quantitative assessment system in which high-quality studies were considered to be those marked "Yes" ten or more times in the case of randomised controlled trial studies, 5 or more times in the case of quasi-experimental studies [25] and 6 or more times in the case of cross-sectional studies [26]. This quality analysis resulted in the elimination of 12 articles.

**2.2.5. Data analysis.** We evaluated the tendencies as regards temporal distribution, journal, main author's listed country, type of autobiographical stimulus and type of emotion evoked. We also analysed the content of the abstracts in order to identify the key topics that have been established in the field of MIPs through the use of autobiographical stimuli.

For the latter, we used text mining techniques, based on the study of keywords, as these offer the possibility of analysing an extensive dataset and exploring the dynamics of the topics within, thanks to a process of deriving patterns and trends from text. Using this approach enabled us to extract hidden relationships based on processing natural language and statistical co-occurrence [27]. Specifically, we used topic modelling, which is a type of probabilistic modelling that reveals significant structures across collections of documents [28], indicating the themes running through a text, as well as latent topics. To this end, once we had selected the articles, we converted them from their original text format to plain text (.txt), selecting only the sections of interest for the study (name of journal, authors, year of publication, title and abstract). All the resulting documents (one per article) were stored in the same folder.

The pre-processing procedure was conducted as follows. First, we eliminated the empty words, or stopwords (e.g., "of", "the", "that", "and", etc.), numbers and punctuation marks. General words, such as "induction", "memory" and "autobiographical" were also omitted to avoid deficient discriminative information, since, given the study aims, the presence of such words was to be expected in almost all the selected texts. Words were set as the unit of analysis. In this stage, we also processed the data using tokenisation (splitting the words into small

pieces, without punctuation marks) and lemmatisation (grouping together inflected words with their lemma or dictionary words) in all the texts, filtering for stopwords. Following this, we converted the lemmatised words to their stems, in order to reduce the number of elements in each text.

Additionally, we performed topic modelling to search and identify groups of words that represented topics (recurrent patterns of co-occurring words) within the dataset. To this end, we used Latent Dirichlet Allocation (LDA) [29] to extract the corpus of the abstract texts. This is a Bayesian hierarchical approach by which the words in the abstract texts were analysed, estimating the distribution of the conjoint probability between what can be observed (the words in the text) and what cannot be observed (the hidden structure of the topics). The LDA was performed following the Gibbs sampling [30]. A matrix had first been created in which the rows represented the text and the columns the words, as the format under which to apply the LDA logarithm. It is worth noting that LDA is an unsupervised method, meaning that before running the model, we have no knowledge of the relationship between the documents. Therefore, we trained the LDA model in accordance with the established data and parameters (words and topics), which yielded a topic distribution matrix with an overall probability of 1. We tested various numbers of topics (3, 4, 5, 6, 7 and 8) to allow for different results and to thus find a balance in terms of an appropriate number of topics. To select the number of topics, we took into account that: a) the terms within each topic had an overall meaning; b) the most representative or most frequently occurring words were directly related to the topics; c) there was no overlap between the different topics; and d) the most important topics within each field of study were included [31].

Using Wordcloud, we conducted a content analysis of the resulting topics, identifying the most common words in each one so as to characterise the individual topics. Additionally, the probabilities of the words in each topic were estimated, using the beta method. Then, using a relational network, we grouped the studies together by topic, following a betweenness centrality criterion that quantified the probability of an article appearing in a particular topic. This was represented as a node within a network, which also showed the connection between topics.

To conduct the analyses, we developed an ad-hoc R script, using the following packages: NLP [32]; tm [33]; topicmodels [34]; gplots [35]; RColorBrewer [36]; reshape2 [37]; easy-PubMed [38]; SnowballC [39]; network [40]; sna [41]; ggplot2 [42]; ggnet [43]; ggrepel [44]; ggraph [45]; wordcloud [46]; tidytext [47]; dplyr [48]; tidyr [49].

The R script is available under request to the corresponding author.

## 3. Results

Below, we present the answers to our research questions:

### RQ1: ¿ How many articles on emotion induction through autobiographical stimuli have been published in the last twenty years?

Fig 2 shows the temporal distribution of the articles on MIPs using autobiographical stimuli published between 2000 and 2021. The median was used to find the distribution of frequency, with 6 being the most common number of publications per year. It can be seen that 2007, 2010, 2013, 2014, 2017 and 2019 are the years with the largest number of publications, 9 in each one. Only one study was published in 2004 and 2005.

The number of publications per year between 2000 and 2006 is low, except for 2003, when 7 articles were published. From 2007, an upward, albeit irregular, trend can be seen. There was a sharp drop in the number of publications in 2018, although over the last three years (2019–

Number of published paper per year

**Fig 2. Publications by year.**

2021), there was an apparent increase, especially if we consider that the data for 2001 only extend as far as June. Despite most of the years included in the study presenting fewer than 6 publications, in recent years, this number has stayed the same or has increased, which supports the notion of a growing trend.

### *RQ2*: Where have the studies on emotion induction interventions using autobiographical stimuli been published? In what areas of psychology and related disciplines can we find studies on emotion induction through autobiographical stimuli?

The studies included in this review were published in a total of 64 journals (56.25% from the field of social sciences and 43.75% from the sciences). Most of the journals (70.31%) have only published one article on the topic under study in this review.

Fig 3 shows that the journal that published the largest number of articles was Emotion (impact factor: 3.177), with a total of 12 (9.52% of the total sample). The other journals with a relatively larger number of articles are *Memory* (impact factor: 1.895) with 11 publications (8.73%), and *Cognition and Emotion* (impact factor: 2.473) and *Journal of Personality and Social Psychology* (impact factor: 6.315) both with a total of 7 publications (5.55%), *Frontiers in Psychology* (impact factor: 2.067) and *Plos One* (impact factor: 2.740) with 5 publications each (3.96%) and *NeuroImage* (impact factor: 5.902), which published 4 articles (3.17%).

All the articles were published in high-quality international journals on research in the field of experimental studies on memory, cognition, emotions, neuropsychology and the treatment of psychopathologies. Of these journals, 26.6% belong to the category of neurosciences (24 articles), 21.9% to the category of experimental psychology (49 articles) and 12.5% can be considered multidisciplinary psychology journals (13 articles). Other categories include social psychology (9.4%), behavioural sciences (7.8%), clinical psychology (6.2%), developmental psychology (3.1%), multidisciplinary sciences (3.1%), geriatrics and gerontology (3.1%), medicine and experimental research (3.1%), psychiatry (1.6%) and computer graphics (1.6%).

*Editing journals of the papers*

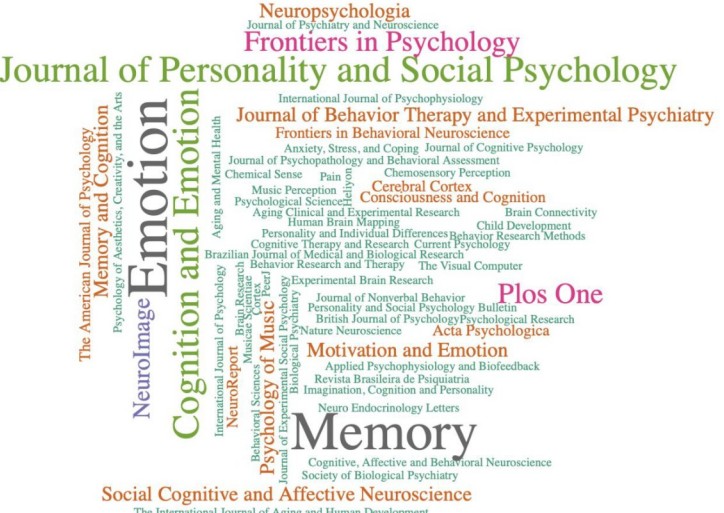

**Fig 3. Publications by journal.**

There is an outstanding presence of specialised journals in research fields related to the neurosciences (e.g., *NeuroImage*, *Social Cognitive and Affective Neuroscience*, *Cerebral Cortex* or *NeuroReport*). These journals focus on the study of mental processes and human behaviour, using methodologies such as functional magnetic resonance imaging, cognitive electrophysiology and brain stimulation (e.g., fMRI, EEG, PET).

The journals specialising in experimental psychology are centred on studying emotional processes (e.g., *Emotion*), memory (e.g., *Memory*), mental processes associated with the emotions (e.g., *Frontiers in Psychology*), the effect of mood state on cognition (e.g. Cognition and Emotion) and emotional self-regulation (e.g. *Motivation and Emotion*).

As regards the quality of the journals and following the impact factor reported in Journal Citation Reports (JCR) for 2019, 33.3% of the journals included in the current review are ranked in Q1, 23.8% in Q2, 28.6% in Q3 and 14.3% in Q4. Thus, it can be said that most of the journals publishing studies on mood induction procedures using autobiographical stimuli have a moderate to high impact factor.

With respect to the country of origin of the main authors, Fig 4 shows that the leading country is the United States, which is home to 42.8% of the institutions to which the main authors of the articles included herein are affiliated. The USA is followed by Germany (9.5%), Canada and Japan (7.1%) and Spain and United Kingdom (5.5%). In short, the scientific output on mood induction procedures using autobiographical stimuli stems from a total of 21 countries, of which 12 are located in Europe.

## RQ3: Is this type of procedure a specialised field?

The studies in this review included 110 different main authors. Fig 5 shows the frequency of publication by lead author, with 88.1% of the authors having only one publication related to the field of study under. A total of 9.1% of the authors have two published articles, while only 2.7% have 3 publications, a number of articles suggesting a certain level of dedication and specialisation in the specific area of study. These authors are Albarracin (University of Florida, USA), Willander (University of Gävle, Sweden) and Denkova (University of Alberta, Canada).

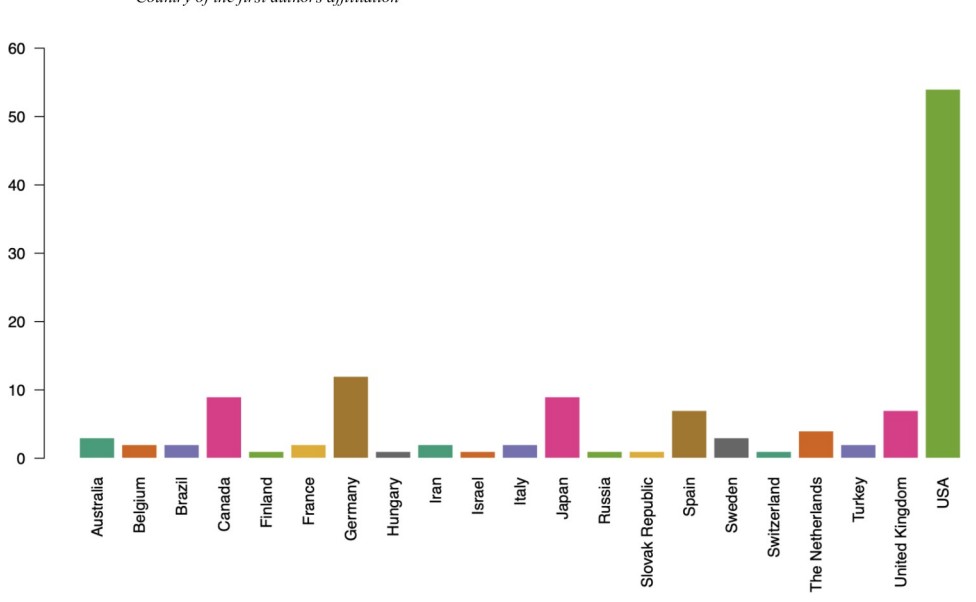

**Fig 4. Publications by country.**

## RQ4: What are the re-occurring topics in the use of autobiographical stimuli for emotion induction?

We extracted the primary topics related to MIPs through autobiographical stimuli. Specifically, four topics were identified, after concluding that extracting a larger number of topics would lead to these being too homogenous, and thus difficult to define and characterise, while a smaller number would lead to a loss of useful information. Fig 6 shows the key words for each of the topics.

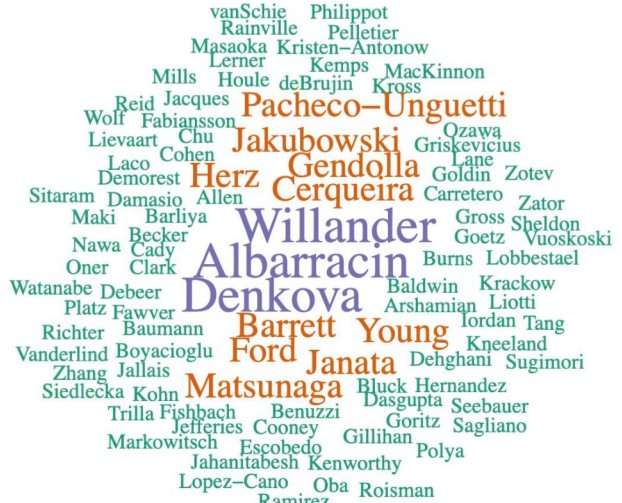

**Fig 5. Publications by authors.**

Topcis

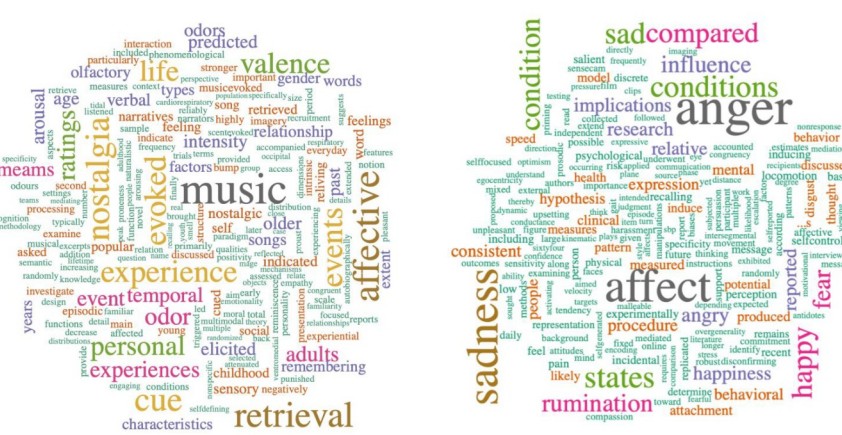

Topic 1 "Cue types"                    Topic 2 "Emotional states"

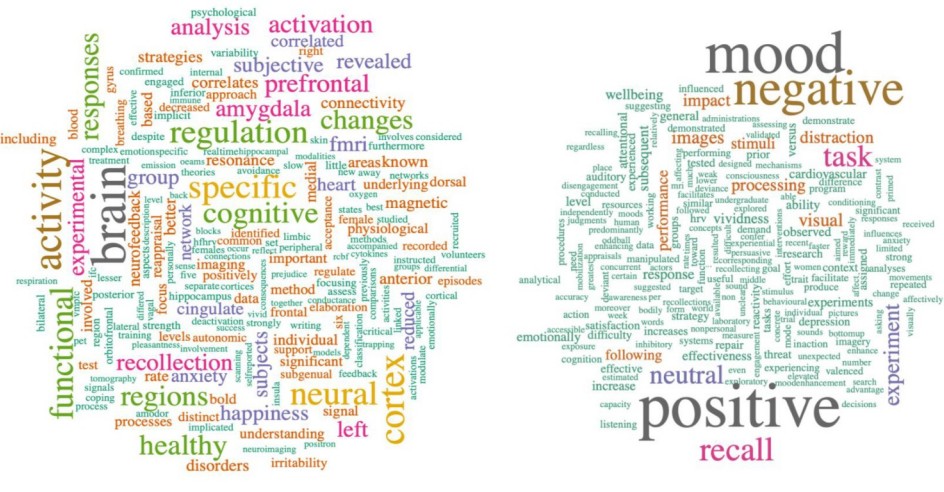

Topic 3 "Neuroimaging and Psychophysiology"        Topic 4 "Induction task"

**Fig 6. Topics.** Topic 1 "Cue types". Topic 2 "Emotional states". Topic 3 "Neuroimaging and Psychophysiology". Topic 4 "Induction task".

The first thing we see is that each of the topics is clearly represented by different words, which suggests that the topics are sufficiently differentiated and thus the information they provide is valuable. A look at the words for each topic allows us to directly and simply visualise the four topics extracted from the overall analysis of the abstracts from all the articles. It is worth noting that some terms can be found in all four topics but with different probabilities, where those with the highest possibilities standing out in the word clouds. Table 1 lists the 20 most frequently occurring words in each.

In Topic 1, *"Cue types"*, we see that the most representative terms refer to the different types of stimuli used to elicit personal memories, with the aim of triggering affective changes. The most widely used cues are musical, visual, olfactory and verbal.

For Topic 2, *"Emotional states"*, the most representative terms are those related to the different mood states induced using autobiographical stimuli procedures, with the most common emotions being anger, sadness, fear and happiness.

**Table 1. Most frequent terms per topic.**

| Topic 1 | Topic 2 | Topic 3 | Topic 4 |
|---------|---------|---------|---------|
| music | anger | brain | positive |
| cue | sadness | activity | mood |
| specific | states | cortex | negative |
| nostalgia | conditions | neural | recall |
| events | sad | functional | affect |
| evoked | condition | responses | task |
| affective | happy | healthy | neutral |
| retrieval | compared | regions | processing |
| experience | happiness | prefrontal | experiment |
| life | research | activation | impact |
| ratings | fear | left | distraction |
| visual | rumination | group | performance |
| experiences | influence | amygdala | stimuli |
| personal | angry | recollection | response |
| images | reported | regulation | following |
| valence | implications | fmri | procedure |
| odor | people | analysis | cognitive |
| adults | relative | subjective | experiments |
| meams | affective | data | predicted |
| event | revealed | changes | recalling |

In Topic 3 "*Neuroimaging and Psychophysiology*", the content is connected to the use of neuroimaging methods in MIPs with autobiographical stimuli to study cognitive functioning and to understand the biological roots of emotions. This topic also reveals references to the analysis of physiological responses produced by the emotional reactions triggered by the AM (e.g., heart rate, skin conductance, blood pressure).

For Topic 4, "*Induction task*", the terms refer to the characteristics of the induction tasks based on memory retrieval through the use of different stimuli designed to observe the effect of the mood states induced (both positive and negative ones). They are related to various aspects, such as emotional processing and the impact on cognitive and behavioural responses and processes.

It can also be seen that the "*Induction task*" topic is more clearly characterised than the others, as it includes a lower number of words with higher probabilities, in contrast to the other three topics, which present a lower variation of probabilities given their similarities.

The relational network (Fig 7) shows the associations between the studies included in this work, according to the topics previously described. In the network, the nodes represent the articles, while the edges determine the distances between the nodes, and thus between articles. Hence, the shorter the edge, the shorter is the distance, indicating a greater relationship in the similarity of the topics in the articles. The size of the nodes represents the numerical variance of the probability of each article belonging to a certain topic, and thus the larger the node, the greater is its probability of belonging to the topic.

Thus, we find that the publications with the highest probability of being associated with the topic of *"Cue types"* (according to abstract content) are P96, P95, P107, P79, P119, P118, P68, P100, P77 and P14. Those with the greatest probability of being associated with the topic of "*Emotional states* are P36, P106, P2, P16, P15, P7, P30, P72 and P73. In the case of *"Neuroimaging and Psychophysiology"*, the articles are P126, P29, P20, P45, P75, P71, P117 and P33, while,

*Relational network*

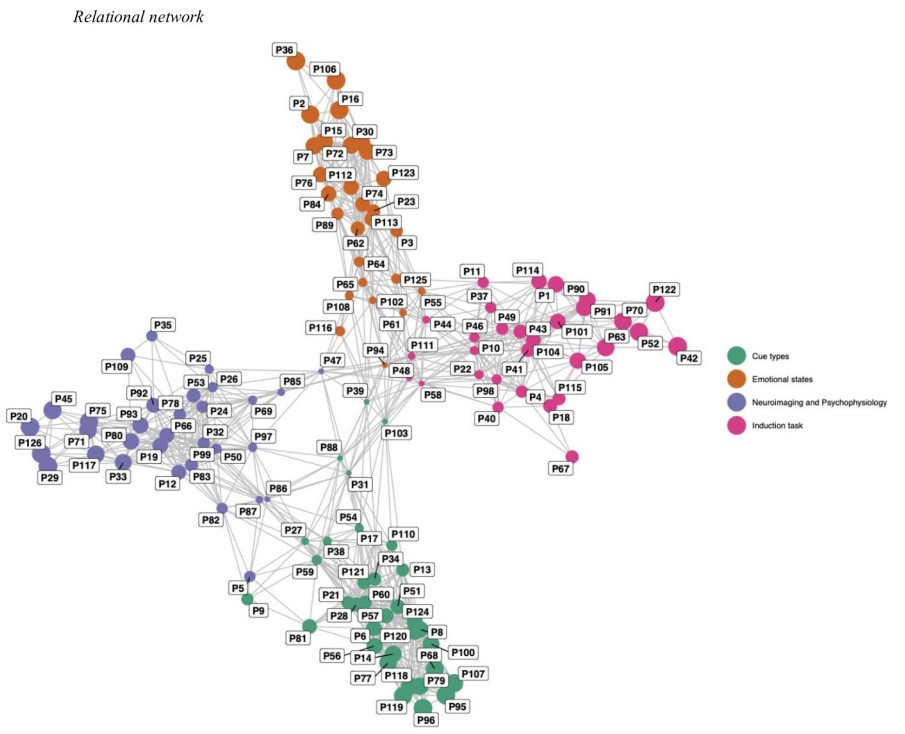

**Fig 7. Article linking.**

finally, for *"Induction task"* the articles with the greatest probability are P42, P122, P52, P70, P63, P90, P91 and P105. The figure shows that the publications with the highest probability of being associated with the "*Emotional states*" topic are more similar to one another compared to the articles corresponding to the other 3 topics, given that the distances between them are shorter.

It is worth noting that the publications within each topic are highly interrelated, especially in the topic of "*Cue types*". It is also noteworthy that some publications, such as P39 o P103, which belong to "*Cue types*" albeit with low probabilities, are also close to the topics of "*Emotional states*" and "*Induction task*" respectively.

## RQ5: What type of autobiographical stimulus is most frequently used in mood induction procedures?

Most (n = 109) studies use only one type of cue, that is, they implement unimodal procedures, while the others (n = 17) use a combination of cues, meaning they apply bimodal or multi-modal methods.

The studies on unimodal MIPs with autobiographical stimuli most frequently deploy verbal cues (74.3%), followed, albeit at a great distance, by musical cues (14.6%), olfactory stimuli (6.5%) and visual cues (4.6%).

The most commonly used bimodal procedures are those combining "verbal + musical" cues, followed by "verbal + olfactory", "verbal + visual", "musical + visual" and "visual + olfactory" cues. Meanwhile, the most frequently used multimodal procedures use the combinations of "musical + visual + olfactory" and "verbal + visual + olfactory" cues.

### RQ6: What types of emotion are most frequently elicited using procedures based on autobiographical memory recall?

Of the studies included in this review, 85.7% use different procedures based on autobiographical memory recall with the aim of eliciting a specific type of emotion, while the remaining 14.3% implement the procedures without the aim of triggering any specific target emotion.

Of the studies specifically designed to induce a concrete type of emotion through the use of autobiographical stimuli, 59.3% seek to induce both positive and negative emotions, 21.3% specifically seek to elicit negative emotions, while 19.4% are intended to specifically induce positive emotions. The most commonly sought negative emotions are sadness, rage, anger, fear and shame, while the most frequently targeted positive emotions are happiness, joy, serenity, pride and nostalgia.

## 4. Discussion

To the best of our knowledge, this is the first article to conduct a systematic mapping review of MIPs based on autobiographical stimuli. Our results may serve to provide researchers with an overview of the current state of the art in the field.

The last twenty years have witnessed an increasing interest in the study of emotions within the disciplines of both psychology and affective neuroscience. This growth has generated new approaches and research questions, as well as new experimental procedures and applications, among which are the manipulation of emotions and the impact on psychological functioning. Accordingly, over recent years, there has been a growing trend in the number of publications on MIPs using autobiographical stimuli, although this growth has been somewhat irregular.

As regards the publication of this type of study, most of the journals in which the articles appear belong to the field of the neurosciences, followed by experimental psychology, suggesting that this is a pioneering field of research grounded in technological innovations. The interest in using such techniques to study the access to AMs stems from its being a complex and demanding process that taps a wide variety of cognitive processes reflected in extensive brain activation patterns [50]. Additionally, the majority of the studies were published in high-impact journals, underlining the high level of scientific evidence they present.

An analysis of the main authors of the studies suggests that the field of research on mood induction through the use of autobiographical stimuli is one that, despite the growing interest, is still not entirely consolidated as there are no researchers with a lengthy tradition in the study of a particular topic. This is reflected by the fact that, most of the 110 different main authors included in the present review have only one publication to their name, with only 3 researchers having published a total of 3 articles.

As regards the number of topics, the 4-topic model provided a good fit and interpretation, identifying: 1) types of cues used in the MIPs based on AM ("Cue types"); 2) emotional states most often elicited in such types of procedures ("Emotional states"); 3) studies conducted in the fields of neuroimaging and psychophysiology ("Neuroimaging and Psychophysiology"); and 4) characteristics of the induction task ("Induction task"). Identifying these topics allowed us to detect research trends in MIPs using autobiographical stimuli.

In the topic of "Emotional states", it can be seen that the most frequently induced emotions are the terms "anger", "sadness", "sad", "happiness", "happy", "fear", "angry". These are studies that seek to induce specific emotions to study their impact on cognitive, behavioural and emotional processes, as, for example, the influence of anger, fear, happiness or sadness on motor behaviour [51]; the effect of anger on pain sensitivity [52]; the effectiveness of happiness as an antidote to sadness [53], the effect of happy and sad mood states on thought confidence [54] or the effect of anger on disconfirming one's own opinion [55].

Regarding "Neuroimaging and Psychophysiology", the most common terms include "brain", "activity", "cortex", "neural" and "functional". This reflects the interest in studying the impact of emotions on cognitive processes and physiological functions through cutting-edge techniques such as the use of functional magnetic resonance imaging (fMRI) [56–58]; real-time functional magnetic resonance imaging (rtfMRI) [59]; electroencephalogram (EGG) signals (EGG) [60]; positron emission tomography (PET) [61]; a well as psychophysiological measures, such as skin conductance response (SCR) and heart rate (HR) [56, 62].

In the "Cue types" topic, the most common terms are "music", "cue", "specific", "nostalgia" and "events". This topic encompasses the most frequently used cues to generate AMs, and, although as previously mentioned, the most widely used cues are verbal, this topic reveals the significant presence of musical stimuli. This greater presence of the term "musical" is because the studies that use verbal cues do not specifically describe the mood induction procedure in their abstracts, while those using musical stimuli emphasise the type of cue used in the procedure [63]. For the same reason, other terms like "visual", "images" and "odour" also occur more frequently than others associated with verbal cues, such as "word".

The most common terms in the topic of "Induction task" are "positive", "mood", "negative", "recall" and "affect", which point to the key characteristics of the mood induction task. In this sense, the tasks tend to be administered individually, with the participants possibly having been assigned to different groups. The procedures entail a variety of cognitive processes (e.g., remembering, reliving, thinking, processing, evaluating, describing, etc.), which serve to assess the emotional impact of the AM participants are asked to retrieve, following the experimenter's instructions.

The most frequently used autobiographical stimuli are unimodal cues, more specifically verbal ones, followed in order by musical cues, olfactory cues, and with a lower frequency, visual stimuli. Under verbal cues, we include all the procedures that seek to elicit AMs using key words (e.g., emotion words, such as "joy", "fear", "love"), verbal instructions and narrative tasks (both spoken and written) based on life stories, in which the participant is asked to recall personal events in which they experienced certain emotions. This technique is commonly used because it is simple to implement in both laboratory and everyday settings, and requires few resources and little time [10].

Musical cues are also frequently used, as music is known to produce and enhance emotions in an individual, facilitating and improving their ability to access autobiographical memories. In other words, the very nature of music leads it to generate emotions and associations with both special and everyday personal events [64]. Additionally, as music has a significant cognitive function related to self-reflection and self-awareness, it is commonly used in the retrieval of events from the past [65, 66]. Musical cues typically include excerpts of popular songs (selected from music charts such as Billboard) associated with the youth life stage, and also childhood, as these are considered to be remembered better and to present high levels of emotionality [67]. Studies also often use excerpts of film soundtracks [68] and pieces of classical music known for their power to induce certain emotional states [4, 69].

Olfactory cues are also popular in MIPs based on AM, as the literature, motivated by the Proust phenomenon, finds that odours evoke memories that are more emotionally loaded, more vivid, specific, rare and relatively older [70]. This is because AM appears to generate strong feelings of reality due to a unique sensory process (associated with the activation of the limbic system) that differentiates olfaction from other sensory modalities [71]. In MIPs using olfactory cues, the odours tend to be familiar, linked to certain life stages, especially childhood [72, 73]. Studies have also used the scent of perfumes selected pre-test by participants due to their power to evoke AMs [62].

As regards visual cues, most studies use images taken from the International Affective Picture System [74, 75], which comprises more than a thousand images related to everyday

situations, rated and classified according to the affective dimension of level of control or dominance, level of arousal or calm and level of pleasantness or unpleasantness [74]. Studies also use photographs designed to represent scenes and objects from concrete life stages (e.g., primary school) [75] or specific events [76], as well as personal photographs [77].

Although most of the studies administer unimodal MIPs, that is, they use only one type of cue, other studies implement two, or three, different types of cues, thus being bimodal or multimodal MIPs. The most common bimodal combinations are those using verbal cues, together with auditory, visual or olfactory stimuli. In these cases, the participant is asked to access and retrieve an autobiographical event from a specific cue that is consistent with the emotion to be induced. Such cues may be auditory [78, 79], visual (e.g. an image) or olfactory [50, 80], with the participant being asked to describe and define the event in written or oral form, or to narrate it in line with a series of guided questions. Of the multimodal combinations (using three or more cues), the most frequent are those exposing participants to musical, visual and olfactory cues [81] and verbal, visual and olfactory cues [82].

Regardless of the stimulus used, MIPs using AM are based on asking a person to engage with their memory, the emotions this entails and to attempt to generate a detailed mental representation of the physiological sensations, feelings and behaviours they experienced during this event from their past [83]. Reliving past events triggers a cascade of emotions that allows a person to connect their present to their past, such that an individual's recalling how they felt in the past may impact on how they feel in the present. It is worth noting that emotions tend to appear in groups, in multiple combinations and varying degrees of intensity [10]. Hence, some studies prefer not to use AM to induce specific, discrete emotions (e.g., fear) given the complexity of emotions evoked through the retrieval of past events, and, as actually occurred in the real-life situations, they allow them to emerge in any type of combination, with no concrete direction. It should also be noted that both positive emotions, such as happiness, and negative emotions, such as sadness, are typically induced using different types of cues, including autobiographical, verbal, olfactory and auditory stimuli. However, emotions such as anger tend to be induced using, above all, verbal cues, whereby the participant is asked to recall a specific event in which they felt a certain emotion [84, 85], but this type of emotion is not commonly generated using cues such as music, odours or images.

Nostalgia (an emotion induced in various publications) may be considered a bittersweet emotion that mixes sadness and joy [86]. It is an emotion that can contain other more basic ones (pride, happiness, love, sadness), but, in the present review, nostalgia has been considered as a positive emotion, since, regardless of the stimulus it is triggered by, its effect on an individual's subsequent states of wellbeing is regarded as essentially positive [87]. Specifically, nostalgia was the target emotion in 6 studies and was especially associated with olfactory stimuli, underlining the robust relationship between nostalgia, olfactory experiences and autobiographical memory [70, 73].

The present review is not without a series of biases and limitations. One of the biases, stemming from the nature of the study itself, is that of the selective notification of the publications. To mitigate this, we conducted the search using four recognised databases, ensuring we could access the most complete list of articles possible. Another possible bias is related to the selection of studies, despite rigorously defining and keeping to the inclusion and exclusion criteria we established. Additionally, we should underline the possible mistakes committed in the data extraction, despite two independent researchers having performed the processes of searching, extracting, evaluating and classifying the information of interest. Nevertheless, it is important to note that the aim of a systematic mapping revision is to provide a broad, general overview of a concrete topic, which may conceal specific underlying aspects.

Given the importance of emotions in people's day-to-day lives and their effect on associated psychological variables, future lines of research that may arise from the evidence allow us to

advance in the study of procedures based on less commonly used autobiographical stimuli, such as photographs. Additionally, we propose investigating different associations of both the type of photo or memory, together with an exhaustive examination of the variable itself, measuring the impact and its consequences on mood. In light of the methodologies analysed, this article lays the groundwork for the importance of establishing a rigorous and clearly defined protocol for better interventions based on mood induction through AM in experimental and natural settings. Furthermore, a new avenue of research would be to conduct meta-analyses that quantitatively reflect the potential of such procedures. Future studies could also be directed in terms of relating the findings of the present review to another specific field of study and context. The methodology applied in this study is well documented and could therefore be replicated on future occasions.

## 5. Conclusions

Emotions are a key part of our everyday lives, with the potential to affect our wellbeing, and, thus, studying and understanding their functioning and impact at both biological and psychological level is essential in interventions. Although there still remain unresolved questions as regards research on emotions and MIPs, the present work establishes a novel theoretical framework on which to base future studies that seek to address the possible causes and consequences of different emotional states on cognitive, behavioural and psychological variables.

Broadly speaking, our findings suggest there has been a growth in the number of studies using MIPs based on autobiographical stimuli, which are underpinned by the high effectiveness of the retrieval of personal memories in activating different. The present study also shows the interest in these types of procedures in the field of the neurosciences, suggesting the emergence of ground-breaking research supported by technological innovations, endowing it with an enhanced level of scientific evidence. There appears to be a preference for the use of verbal cues, followed by musical, olfactory and visual stimuli, for inducing previously defined specific emotions, of both positive and negative valence. Accordingly, future studies should be conducted to compare the efficacy of different types of cues in generating specific types of emotions.

## Supporting information

**S1 Checklist.**
(PDF)

**S1 File. Checklist: Template for a mapping study protocol.**
(DOCX)

**S2 File. Bibliography of the studies included in the review.**
(DOCX)

**S3 File. Extraction of relevant data.**
(DOCX)

## Acknowledgments

We would like to thank all the people who have selflessly collaborated in this article.

## Author Contributions

**Conceptualization:** Dolores Fernández-Pérez, Abel Toledano-González, Laura Ros, José M. Latorre.

**Methodology:** Laura Ros, José M. Latorre.

**Writing – original draft:** Dolores Fernández-Pérez, Abel Toledano-González, Laura Ros, José M. Latorre.

**Writing – review & editing:** Dolores Fernández-Pérez, Abel Toledano-González, Laura Ros, José M. Latorre.

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
