## [Decision Letter · Decision Letter 0]

17 Feb 2022

PONE-D-21-39680Use of autobiographical stimuli as a mood manipulation procedure: Systematic Mapping ReviewPLOS ONE

Dear Dr. Toledano-González,

Thank you for submitting your manuscript to PLOS ONE. After careful consideration, we feel that it has merit but does not fully meet PLOS ONE’s publication criteria as it currently stands. Therefore, we invite you to submit a revised version of the manuscript that addresses the points raised during the review process.

 Please submit your revised manuscript by Apr 03 2022 11:59PM. If you will need more time than this to complete your revisions, please reply to this message or contact the journal office at plosone@plos.org. Please include the following items when submitting your revised manuscript:A rebuttal letter that responds to each point raised by the academic editor and reviewer(s). You should upload this letter as a separate file labeled 'Response to Reviewers'.A marked-up copy of your manuscript that highlights changes made to the original version. You should upload this as a separate file labeled 'Revised Manuscript with Track Changes'.An unmarked version of your revised paper without tracked changes. You should upload this as a separate file labeled 'Manuscript'.

We look forward to receiving your revised manuscript.

Kind regards,

César Leal-Costa, Ph. D

Academic Editor

PLOS ONE

Journal Requirements:

“The authors received funding from the European Regional Development Funds (FEDER) [2018/11744] [2019/7375] [2020/3771].

This work was supported by the Castilla-La Mancha Department of Education, Culture and Sports and the European Regional Development Fund under SPBLY/19/180501/000181 grant and the Spanish Ministry of Science, Innovation and Universities under PID2019-103956RB-I00 grant.”

4. We note that this manuscript is a systematic review or meta-analysis; our author guidelines therefore require that you use PRISMA guidance to help improve reporting quality of this type of study. Please upload copies of the completed PRISMA checklist as Supporting Information with a file name “PRISMA checklist”.

Reviewers' comments:

Reviewer's Responses to Questions

**Comments to the Author**

1. Is the manuscript technically sound, and do the data support the conclusions?

Reviewer #1: Yes

Reviewer #2: Yes

2. Has the statistical analysis been performed appropriately and rigorously? 

Reviewer #1: N/A

Reviewer #2: Yes

3. Have the authors made all data underlying the findings in their manuscript fully available?

Reviewer #1: Yes

Reviewer #2: Yes

4. Is the manuscript presented in an intelligible fashion and written in standard English?

Reviewer #1: Yes

Reviewer #2: Yes

5. Review Comments to the Author

Reviewer #1: Please, include script file. This information will help readers to understand the process of your study.

Please, add this reference:

Jordan, Zoe PhD; Lockwood, Craig PhD; Munn, Zachary PhD; Aromataris, Edoardo PhD The updated Joanna Briggs Institute Model of Evidence-Based Healthcare, International Journal of Evidence-Based Healthcare: March 2019 - Volume 17 - Issue 1 - p 58-71 doi: 10.1097/XEB.0000000000000155

please improve tool description

Pubmed is not a database. PubMed is a free search engine accessing primarily the MEDLINE database

Could you add a wordcloud (or tag cloud)?

Reviewer #2: This manuscript addresses an interesting and important issue in psychological research: Use of autobiographical stimuli as a mood manipulation procedure. The literature can benefit from studies that clarify the mood induction procedures.

My comments are organized according to the sections of the paper, with the most important issues in each section addressed first. The following are some concerns and recommendations for strengthening this paper.

Abstract: It would be useful to include a broad description of the conclusions in the abstract.

Keywords. “Autobiographical” keyword is redundant with the title. Perhaps, authors could include another keyword

Introduction

The introduction should include some aspect that describes why interest in the study of use of autobiographical stimuli as a mood manipulation procedure has increased (or the benefits of its increase) and what gaps this review intends to cover.

Materials & Methods

Authors follow a systematic mapping review approach. It would be rigorous to include some reference that justifies its suitability for the objectives set forth in this work.

Although one of the inclusion criteria is " Articles not published in English " it would be useful to include Spanish written articles since the reviewers are from Spain

Authors state on page 6 that “Two researchers independently reviewed and filtered all 3296 articles, reading and analysing the titles, abstract and keywords and following the inclusion and exclusion criteria. When the two researchers were unable to agree, a third researcher independently resolved the conflict.”

Nevertheless authors state on page 7 that “Data extraction was carried out by two researchers (DFP and ATG). Any discrepancies were resolved by two independent researchers (JML and LR).”

Are one or two researchers resolving the conflicts?

Results

Authors state on page 10 that “Of these journals, 26.6% belong to the category of neurosciences (24 articles), 21.9% to the category of experimental psychology (49 articles) and 12.5% can be considered multidisciplinary psychology journals (13 articles). Other categories include social psychology (9.4%), behavioural sciences (7.8%), clinical psychology (6.2%), developmental psychology (3.1%), multidisciplinary sciences (3.1%), geriatrics and gerontology (3.1%), medicine and experimental research (3.1%), psychiatry (1.6%) and computer graphics (1.6%).”

What criteria was used? How do authors choose if a journal belongs to several categories?

Discussion

Authors state on page19 that “To mitigate this, we conducted the search using three recognised databases”.

Nevertheless they have analysed four recognised databases (Scopus, Pubmed, PsycINFO and Web of Science).

Have the analysed three or four databases?

Bibliography

Please include the Journal name abbreviation (Adv Exp Soc Psychol) in the following reference:

Sedikides C, Wildschut T, Routledge C, Arndt J, Hepper EG, Zhou X. To nostalgize: Mixing memory with affect and desire. Advances in Experimental Social Psychology. 2015. 189–273. doi:10.1016/bs.aesp.2014.10.001

Although it is Optional, It would be nice if you translate the article title (MEDLINE/PubMed practice) in your non English referece:

Ricarte JJ, Latorre JM, Ros L. Diseño y análisis del funcionamiento del Test de Memoria Autobiográfica en población española. Apunt Psicol. 2013;31:3–10.

Figures

The PRISMA Diagram could be improved by indicating the reasons ( and the number) that are excluded at each step

Supplementary material

I wonder if English names for the S1 and S3 files attached as supplementary materials would be more appropriate than Spanish names.

Material suplementario S1.docx

Material suplementario S3.docx

Recommendation: accept with minor changes, because despite these limitations, few previous studies have explored the autobiographical stimuli as a mood manipulation procedure.

6. PLOS authors have the option to publish the peer review history of their article (what does this mean?). If published, this will include your full peer review and any attached files.

Reviewer #1: No

Reviewer #2: No

---

## [Author Response · Author response to Decision Letter 0]

22 Mar 2022

In the attached files you can find the required modifications. 

The funders had no role in the study design, data collection and analysis, decision to publish, or preparation of the manuscript.

---

## [Decision Letter · Decision Letter 1]

1 May 2022

PONE-D-21-39680R1Use of autobiographical stimuli as a mood manipulation procedure: Systematic Mapping ReviewPLOS ONE

Dear Dr. Toledano-González,

Thank you for submitting your manuscript to PLOS ONE. After careful consideration, we feel that it has merit but does not fully meet PLOS ONE’s publication criteria as it currently stands. Therefore, we invite you to submit a revised version of the manuscript that addresses the points raised during the review process.

One of the reviewers has made minor comments.  Please respond to his comments that he has put in the attached document.  After that review I will make the final decision.

We look forward to receiving your revised manuscript.

Kind regards,

César Leal-Costa, Ph. D

Academic Editor

PLOS ONE

Journal Requirements:

Reviewers' comments:

Reviewer's Responses to Questions

**Comments to the Author**

1. If the authors have adequately addressed your comments raised in a previous round of review and you feel that this manuscript is now acceptable for publication, you may indicate that here to bypass the “Comments to the Author” section, enter your conflict of interest statement in the “Confidential to Editor” section, and submit your "Accept" recommendation.

Reviewer #1: All comments have been addressed

Reviewer #2: All comments have been addressed

2. Is the manuscript technically sound, and do the data support the conclusions?

Reviewer #1: Yes

Reviewer #2: Yes

3. Has the statistical analysis been performed appropriately and rigorously? 

Reviewer #1: N/A

Reviewer #2: Yes

4. Have the authors made all data underlying the findings in their manuscript fully available?

Reviewer #1: Yes

Reviewer #2: Yes

5. Is the manuscript presented in an intelligible fashion and written in standard English?

Reviewer #1: Yes

Reviewer #2: Yes

6. Review Comments to the Author

Reviewer #1: This is an interesting study that the authors exploring the Use of autobiographical stimuli as a mood manipulation procedure. This is a relevant and interesting article, I suggest its publications.

Reviewer #2: This manuscript addresses an interesting and important issue in psychological research: Use of autobiographical stimuli as a mood manipulation procedure. The authors have adequately addressed the comments and concerns I raised in my initial review. The manuscript has been significantly improved and now warrants publication Plos One:

a.- Authors have expanded the Conclusions section of the abstract to include what they see primary findings of the study

b.- Authors have changed “Autobiographical” by “personal memories” in the keywords section

c.- The revised manuscript’s Introduction highlights the importance of the interest in the study of autobiographical stimulus as a mood induction procedure

d.- Manuscript has included a reference that justifies the use of a systematic mapping review

e.- Authors now state that the classifications used by category of the journals that contained publications included in their article were extracted directly from Web of Sciences (Clarivate) under "Journal Information" and "JCR Category".

f.- Manuscript includes that final articles were extracted from 4 recognized databases (Scopus, Pubmed (MEDLINE), PsycINFO and Web of Sciences).

Recommendation: I think that from a qualitative viewpoint, the manuscript is ready for publication. I congratulate the authors on this useful contribution to the literature focusing on mood manipulation procedure.

7. PLOS authors have the option to publish the peer review history of their article (what does this mean?). If published, this will include your full peer review and any attached files.

Reviewer #1: No

Reviewer #2: No

---

## [Author Response · Author response to Decision Letter 1]

11 May 2022

The response letter to the reviewers is uploaded as a file in "Attach Files". 

I also attach in this space all the comments and changes made on the document to facilitate this task as much as possible. 

Response letter to reviewers.

Ms. Ref. No.: PONE-D-21-39680

Original Title: Use of autobiographical stimuli as a mood manipulation procedure: Systematic Mapping Review

PLOS ONE

Dear editor and reviewers,

Thank you once again for your time and dedication towards this article in making the following suggestions detailed below. 

To facilitate the task of the reviewers and editors, the changes made will be detailed following the outline provided. 

Kind regards.

Reviewers' comments:

This is an interesting study that the authors exploring the Use of autobiographical stimuli as a mood manipulation procedure. This is a relevant and interesting article. However, I would like to suggest to the authors certain aspects to improve

REPLY: Thank you very much for your comment and perspective on our research article.

Please change, Pubmed (MEDLINE) to Medline (PubMed). 

REPLY: The changes indicated by the reviewers have been carried out by changing the order of the words.

Please dele the space the word “article” and the dot. See: we analysed a total of 126 published articles. 

REPLY: The changes have been carried out by eliminating the marked word together with the period, connecting both sentences.

Please dele the space the word “topics” and the dot. See: Text mining techniques were used to extract the main related topics.

REPLY: The space, word and period have been removed from the marked sentence to modify it.

Keywords should place in alphabetical order.

REPLY: The keywords are sorted alphabetically

Please add a reference to the first sentence “In recent years, there has been growing interest in studying emotions and their impact on different psychological processes and characteristics.”

REPLY: Two references have been included in the text showing the scientific interest of other researchers on the subject. 

1. Luminet O. (2022). Towards a better integration of emotional factors in autobiographical memory. Memory, 30(1), 49-54. doi: 10.1080/09658211.2021.1896738

2. Sheldon S, Chu S, Nitschke JP, Pruessner JC, Bartz JA. The dynamic interplay between acute psychosocial stress, emotion and autobiographical memory. Sci Rep. 2018;8, 8684. doi: 10.1038/s41598-018-26890-8

Please change, PubMed (MEDLINE) to Medline (PubMed). 

REPLY: The changes indicated by the reviewers have been carried out by changing the order of the words.

Please add an example of the search strategy.

REPLY: The search string used in the databases Scopus (Elsevier), Medline (PubMed), PsycINFO (American Psychological Association) to obtain all the articles included in the figure was as follows: ("Autobiographical Memory") AND ("Emotional Induction" OR "Emotion Regulation" OR Emotion") in the title, abstract and keywords fields for their final result. . This string of words was used in each of them by selecting articles between the years 2000 and 2021.

Any discrepancies should be resolved with a third researcher.

REPLY: This has been modified in the text. All discrepancies that arose during article filtering were resolved by a third reviewer.

Please add future implications.

REPLY: In the final part of the discussion, we add the future implications based on the evidence shown in the resulting articles: 

“Given the importance of emotions in people’s day-to-day lives and their effect on associated psychological variables, future lines of research that may arise from the evidence allow us to advance in the study of procedures based on less commonly used autobiographical stimuli, such as photographs. Additionally, we propose investigating different associations of both the type of photo or memory, together with an exhaustive examination of the variable itself, measuring the impact and its consequences on mood. In light of the methodologies analysed, this article lays the groundwork for the importance of establishing a rigorous and clearly defined protocol for better interventions based on mood induction through AM in experimental and natural settings. Furthermore, a new avenue of research would be to conduct meta-analyses that quantitatively reflect the potential of such procedures. Future studies could also be directed in terms of relating the findings of the present review to another specific field of study and context. The methodology applied in this study is well documented and could therefore be replicated on future occasions.”

---

## [Editor Report · Decision Letter 2]

20 May 2022

Use of autobiographical stimuli as a mood manipulation procedure: Systematic Mapping Review

PONE-D-21-39680R2

Dear Dr. Toledano-González,

We’re pleased to inform you that your manuscript has been judged scientifically suitable for publication and will be formally accepted for publication once it meets all outstanding technical requirements.

Kind regards,

César Leal-Costa, Ph. D

Academic Editor

PLOS ONE
---

## [Editor Report · Acceptance letter]

15 Jun 2022

PONE-D-21-39680R2 

Use of autobiographical stimuli as a mood manipulation procedure: Systematic Mapping Review 

Dear Dr. Toledano-González:

I'm pleased to inform you that your manuscript has been deemed suitable for publication in PLOS ONE. Congratulations! Your manuscript is now with our production department. 

Kind regards, 

on behalf of

Dr. César Leal-Costa 

Academic Editor

PLOS ONE